# Fetal Hemoglobin in Sickle Hemoglobinopathies: High HbF Genotypes and Phenotypes

**DOI:** 10.3390/jcm9113782

**Published:** 2020-11-23

**Authors:** Martin H. Steinberg

**Affiliations:** Department of Medicine, Division of Hematology/Oncology, Center of Excellence for Sickle Cell Disease, Center for Regenerative Medicine, Genome Science Institute, Boston University School of Medicine and Boston Medical Center, 72 E. Concord St., Boston, MA 02118, USA; mhsteinb@bu.edu

**Keywords:** hereditary persistence of fetal hemoglobin, gamma-globin gene, gene deletion, globin gene expression, genotype/phenotype

## Abstract

Fetal hemoglobin (HbF) usually consists of 4 to 10% of total hemoglobin in adults of African descent with sickle cell anemia. Rarely, their HbF levels reach more than 30%. High HbF levels are sometimes a result of β-globin gene deletions or point mutations in the promoters of the HbF genes. Collectively, the phenotype caused by these mutations is called hereditary persistence of fetal hemoglobin, or HPFH. The pancellularity of HbF associated with these mutations inhibits sickle hemoglobin polymerization in most sickle erythrocytes so that these patients usually have inconsequential hemolysis and few, if any, vasoocclusive complications. Unusually high HbF can also be associated with variants of the major repressors of the *HbF* genes, *BCL11A* and *MYB*. Perhaps most often, we lack an explanation for very high HbF levels in sickle cell anemia.

## 1. Fetal Hemoglobin Levels in Sickle Cell Anemia

Fetal hemoglobin (HbF; α_2_γ_2_), encoded by two nearly identical γ-globin genes (*HBG2*, *HBG1*) that are part of the β-globin gene (*HBB*) cluster (11p15.4), comprises 70 to 90% of the hemolysate in newborns, falling to <1% after 12 months [1]. The “switch” from fetal to adult hemoglobin synthesis in sickle cell anemia (homozygosity for the sickle hemoglobin gene) takes 5 to 10 years and is rarely complete. The sickle hemoglobin gene (*HbS*) is found on five common genetic backgrounds, or haplotypes. Patients of African descent have Bantu, Benin, Cameroon or Senegal haplotypes; in most of these patients a stable HbF level of ~4 to 10% is reached at ~5 years of age. In individuals of Middle Eastern or Indian extract with the Arab–Indian haplotype, stable HbF levels of ~17% are achieved at ~10 years of age [2,3]. Sometimes, in treatment-naive adults of African descent HbF levels between 15 and 30% are present; this can be accompanied by mild disease [4]. Patients with the Arab–Indian haplotype can have HbF far above 20%, but no informative genetic studies of the very high HbF phenotype have been done in this population; the following sections deal only with patients of African ancestry [5]. Among the causes of unusually high HbF levels are deletions or single base substitutions in the *HBB* gene cluster and single nucleotide polymorphisms (SNPs) in the genes *BCL11A* and *MYB* that encode repressors of *HbF* gene expression. This review focuses on the genetic basis and clinical implications of unusually high HbF levels in sickle cell anemia patients of African descent. Comprehensive reviews discuss more general aspects of HbF in sickle cell anemia [6,7,8,9,10,11]. It is worth remembering that newborn screening for sickle cell anemia by fractionation of the hemoglobin proteins cannot distinguish amongst causes of the very high HbF phenotypes. This requires clinical follow-up, hematologic studies, and, to understand the molecular basis of these phenotypes, detailed genetic studies. 

## 2. Persistent High HbF: Mutations of the β-globin Gene Cluster

### 2.1. Hereditary Persistence of Fetal Hemoglobin and δβ Thalassemia

Hereditary persistence of fetal hemoglobin, or HPFH, a term perhaps first coined in 1958, is defined by deletions or point mutations within the *HBB* gene cluster [12]. Large deletions that result in higher than normal levels of HbF persisting into adulthood cause the most frequently recognized type of HPFH (Figure 1). δβ thalassemia is also caused by *HBB* gene cluster deletions but has some hematologic features of thalassemia. HPFH and δβ thalassemia are overlapping phenotypes; their distinction is historical and semantic. The sizes of the causative deletions overlap while their 5′ and 3′ breakpoints differ. Deletions of different sizes with diverse breakpoints might differentially change the interactions among γ-globin genes and their upstream and downstream regulatory elements, such as the locus control region (LCR), and transcription factor complexes. This could account for the wide variation in HbF levels in heterozygous carriers of these mutations and in compound heterozygotes with the *HbS* gene. Identical HPFH-causing mutations, can be associated with dissimilar HbF levels. Although the causes of this are unknown, this might be a result of the inheritance of the minor alleles of the quantitative trait loci (QTL) modulating HbF production that are associated with higher HbF and are discussed below. Reports of QTL genotypes in HPFH have not been published. 

Pancellular vs. heterocellular HbF distribution has been used as a defining aspect of HPFH. All erythroid precursors have the same mutation so that this distinction could simply be an artifact of the insensitivity of HbF measurement in individual erythrocytes. In pancellular HPFH, whether measured either by fluorescence-activated cell sorting (FACS) or Betke staining, the concentration of HbF among F-cells, or cells containing measurable HbF, is not homogeneous. In most studies ~6 pg of HbF/erythrocyte is required for the detection of an F-cell by the commonly used FACS assay. The pancellularity of HbF becomes obvious with ~30% HbF in HbS-HPFH where red cells can have an average of 10 pg of HbF/F-cell, well above the limits of FACS detection. In HPFH due to *HBG2*/*HBG1* promoter mutations where HbF levels can be 10% or less, pancellularity would result in HbF concentrations of 3 to 4 pg of HbF/F-cell, a concentration well below the limits of FACS detection. Therefore, HbF appears pancellular when HbF levels are high and heterocellular when HbF is lower. Differences between pancellular and heterocellular HPFH is likely to reflect the magnitude of the increase in *HBG2* or *HBG1* expression caused by HPFH mutations. Heterocellular HbF distribution might also be a result of epigenetic influences on gene expression leading to position-effect variegation, as proposed by Wood [13,14]. With this context, let us now consider the persistent high HbF caused by mutations in the β-globin gene cluster. 

### 2.2. HbS-Gene Deletion HPFH and δβ Thalassemia

Deletions of about 85 kilobases that remove *HBB* and *HBD* account for the most commonly reported HPFH 1 and HPFH 2 mutations although many other HPFH deletions have been recognized [13]. In HPFH 1 and 2, *HBG2* and *HBG1* are preserved and HbF levels of ~30% are present in heterozygotes; homozygotes have 100% HbF. Based on hematologic and electrophoretic findings, about 0.1% of African Americans were estimated to have the HPFH genotype; an estimation of 1 case in 14,000 was made from the Jamaican birth cohort [15,16,17]. The nearly uniform presence of mild microcytosis suggests that compensatory increase in γ-globin chain (HBG2/HBG1) production is nearly, but not totally, complete even though balanced α- to non-α-globin synthesis has been found [18]. All erythrocytes contain some HbF. This pancellular distribution was first pointed out by Conley in 1963, who attributed the benignity of HbS-HPFH to this HbF distribution [16]. When HbS is present in compound heterozygosity with an HPFH deletion, HbS polymerization is inhibited in most erythrocytes resulting in nearly normal hematology; ^51^Cr red cell half-life was normal (Table 1) [16,19]. A problem in understanding the phenotype of HbS-HPFH is that the hemoglobin genotypes are not always precisely defined. In the largest series of molecularly defined HbS-HPFH, where gap-PCR assays were designed specifically to detect HPFH 1 and HPFH 2, HbF was 31.3 ± 2.4% in 30 cases aged >5 year [19].

The clinical phenotype, while usually referred to as “benign” is less definitively characterized than the hematologic phenotype because of the rarity of HbS-HPFH and the lack of prospective case series with uniform and detailed clinical information. Additionally, most reported patients are young, there is a strong inverse relationship between HbF and age and certain complications of sickle cell disease are age-related [19]. Cases have been reported of osteonecrosis, joint pain, splenomegaly, splenic infarction, retinopathy and hemiparesis. In most instances the *HBB* mutation was not characterized so the cause of the HPFH phenotype was undefined [16,20]. Of 13 adults with HbF > 14%, who were not taking hydroxyurea, which increases MCV and HbF, one individual was deemed to have HbS-HPFH; she had painful episodes. This 25-y-old woman had HbF of 28.5% that appeared pancellularly distributed; hemoglobin concentration was 11.7 g/dL, LDH 145 IU/L and bilirubin 0.6 mg/dL. The MCV of 89.4 fL was unusually high for the HbS-HPFH genotype. The molecular basis of HPFH was not described in detail [21]. Two patients with HbS-HPFH had a history of pain episodes and transfusion. Both had the Indian HPFH 3 that removes 48.5 kb of DNA, starting from the 5′ end of the ψβ gene to a region 30 kb downstream of *HBB* [22,23]. Their HbF was 27 and 29%, hemoglobin 12.0 and 11.5 g/dL and MCV 74.5 and 72.8 fL. Thirteen Brazilian children with HbS-HPFH were studied using gap-PCR, nine had HPFH 2 deletion and four had HPFH 1. Hemoglobin concentration was 12.5 ±  0.6 g/dL with mild microcytosis and HbF was 42.3%  ±  2% (this HbF level, which is much higher than usually reported in HbS-HPFH, was quantified by alkaline electrophoresis that is not the preferred means of analysis). Acute chest syndrome and painful episodes occurred in four children; nine were asymptomatic; none had abnormal transcranial Doppler exams [24]. Splenic infarction was reported in an 18 year old man with HbS-HPFH 2 [25].

Hematologic and family studies of individuals with HbF levels > 20% were used to discriminate among 13 patients with HbS-HPFH, four with HbS-β^0^ thalassemia and 10 with sickle cell anemia [26]. HbA_2_ levels were high in HbS-β^0^ thalassemia. Compared with sickle cell anemia, HbS-HPFH had higher HbF and hemoglobin concentrations; MCV, reticulocyte counts and bilirubin levels were lower. Although the molecular basis of neither HPFH nor β thalassemia was determined, in this and many other studies it is usually possible to ascertain the presence of HbS-deletion HPFH by a combination of HbF measurement, blood counts, erythrocyte indices and hemoglobin phenotyping in informative family members. While this is sufficient for patient management, DNA-based ascertainment of the hemoglobin genotype should be used if reproductive counseling is provided.

δβ thalassemia is also caused by different deletions in the *HBB* gene cluster. These deletions remove *HBB*, *HBD* and sometimes *HBG1*. In the latter case only HBG2 is present (^G^γ (^A^γδβ)^0^ thalassemia). HbS-δβ thalassemia deletions sparing both γ-globin genes (^G^γ^A^γ (δβ)^0^ thalassemia) tend to be 10–20 kb long compared with ~85 kb in HPFH 1 and 2 [13]. Although ^G^γ^A^γ (δβ)^0^ thalassemia is a common cause of β^0^ thalassemia in some Mediterranean populations, there are few well documented examples of compound heterozygosity with the HbS gene. In δβ thalassemia, HbF distribution is said to be heterocellular. The compensation for absent β-globin synthesis in δβ thalassemia is usually less complete than in the common HPFH deletions accounting for lower hemoglobin levels. The phenotype of HbS-δβ thalassemia is difficult to define precisely but might be similar to that of HbS-HPFH.

### 2.3. HbS-HPFH Due to Point Mutations

This genotype, the result of many different mutations in the promoters of *HBG2* and *HBG1* and a single C > T polymorphism in *HBG2*, is associated with a diversity of HbF levels (Table 2). The point mutation can be in *cis* or in *trans* to the HbS gene [27]. Once referred to as heterocellular HPFH, HbF can be distributed both pancellularly in cases where HbF levels are high or heterocellularly when HbF is lower, a likely reflection of the level of *HBG2* or *HBG1* expression [4]. Promoter mutations and the *HBG2* polymorphism (rs7482144) increase HbF as a result of altered binding of the major HbF repressors, ZBTB7A and BCL11A, to their binding motifs and perhaps through the recruitment of other transcriptional activators like GATA1 to their nearby binding motifs. The most important HbF suppressor motifs lie ~200 and ~115 base-pairs upstream of the transcriptional start sites of each γ-globin gene and bind ZBTB7A and BCL11A, respectively (Table 2) [28,29]. Disruption of the *HBG2* QTL 158 bps upstream of this gene and marked by rs7482144, increased HbF about half as much as disruption of the ZBTB7A and BCL11A binding motifs. Rs7482144 affects only *HBG2*, whereas both γ-globin genes are modulated by ZBTB7A and BCL11A, perhaps accounting for this difference [30,31]. The diversity of HbF levels associated with point mutations at the same nucleotide or adjacent nucleotides might be explained by how the physical interactions between the transcription factor and its binding motifs or interactions among the proteins in a transcription factor complex are perturbed. 

Both a point mutation and rs7482144, the −158 C > T polymorphism, can be in cis. This confluence could possibly account for the unusually high HbF in two HbS heterozygotes that had the −158 C > T polymorphism and the −175 T > C mutation and had HbF of 27 and 41% [32,33]. The C > G mutation −202 bp upstream of *HBG2* was found in a man with HbSC disease who had a hemoglobin of 15.4 g/dL, MCV 86 fL, 55% HbS, 18% HbC and 25% HbF. The mutation was presumed to be in *cis* to HbC [33]. 

Point mutations in transcription factor binding sites in patients with sickle cell anemia or other hemoglobin disorders are rarely detected, especially when HbF levels are not high enough to prompt further studies. The phenotype of sickle cell anemia with γ-globin gene promotor mutations resembles that of HbS-deletion HPFH if HbF levels are ~30% and distributed pancellularly. Nevertheless, the number of cases reported is far fewer than reports of HbS-deletion HPFH. 

## 3. Quantitative Trait Loci Modulating HbF Production

Most patients with HbS and very high levels of HbF do not have any of the aforementioned HPFH mutations and continue to have hemolytic anemia and sickle vasoocclusive complications, albeit perhaps at a lower rate. This is likely to be a result of the extreme heterogeneity of HbF distribution amongst F-cells, where despite the high total HbF level, a clinically important fraction of erythrocytes contain insufficient HbF to protect them from HbS polymer inflicted damage [4]. Perhaps interpatient variability in erythropoietic stress accounts for very high HbF levels in some individuals lacking HPFH mutations. Studies in primates suggested wide inter-animal differences in the HbF response to stress erythropoiesis [37]. 

Recent reviews have summarized the molecular basis of HbF repression in the fetus and the ensuing switch to adult hemoglobins [38,39,40]. Higher than expected levels of HbF in sickle cell anemia can be a result of polymorphisms of the major QTL that modulate HbF gene expression. Three QTL, *BCL11A*, *MYB* and *HBG2*, are known to be associated with HbF levels accounting for 10 to 50% of HbF variance [41]. The minor allele frequencies of the sentinel variants, or single nucleotide polymorphisms (SNPs) of these QTL, in sickle cell anemia varies by geographic origin. In sickle cell anemia, the sentinel SNP of the *HBG2* QTL, rs7482144 is present only in the Senegal haplotype of the HbS gene common in patients from West Central Africa and in patients who have the Arab-Indian HbS haplotype and ancestry from Eastern Saudi Arabia or India. 

From the records of the Hemoglobin Diagnostic Reference Laboratory (www.bu.edu/sicklecell), Akinsheye and her coworkers culled samples from 20 African Americans who were homozygous for the HbS gene with HbF of 17.2 ± 4.8 g/dL, were more than five years old, had MCV < 100 fL and who had neither gene deletion HPFH nor point mutations in HbF gene promoters. These individuals were compared with 30 similar patients with HbF of 5.0 ± 2.5% [34]. Minor alleles of the sentinel SNPs of *BCL11A* (rs766432) and *MYB* (rs9399137) were more frequent in the high HbF group and accounted for 20% of HbF variance. The number of minor alleles present (NOMAP score) [42] in the high HbF patients was significantly greater than in the low HbF controls. These results were replicated when 56 HbS homozygotes with HbF of 20.7 ± 8.2%, who were followed in the Cooperative Study of Sickle Cell Disease (CSSCD), were compared with 489 individuals with HbF of 3.1 ± 1.5%. The minor allele frequencies of *BCL11A*, *MYB* and *HBG2* were all significantly higher in the high HbF group. Primate studies have suggested that erythropoietic stress increases γ-globin gene expression when *BCL11A* was downregulated using CRISPR-Cas9 editing of its erythroid enhancer [43].

## 4. Therapeutic Targeting HbF QTL and Mutations

Understanding the mechanisms of some instances of unusually high HbF in sickle cell anemia and knowing the clinical benefits associated with high pancellular levels of HbF is being exploited for possible therapeutic benefit.

The ~13 kb Sicilian HPFH deletion was created in CD34^+^ hematopoietic progenitor cells using CRISPR-Cas9 in an attempt to reproduce the high HbF associated with this deletion [44]. In erythroid colonies derived from cells homozygous for this deletion, there was no *HBB* mRNA; cells heterozygous for the deletion synthesized half as much *HBB* mRNA as control cells. The ratio of *HBG/HBB+HBG* mRNA was ~0.1 in controls, ~0.3 in deletion heterozygotes and 1 in deletion homozygotes. Based on these studies, to be applicable therapeutically, only cells heterozygous for the deletion are needed. 

CRISPR-Cas9 was used to mutate the 13 bp sequence in the promoters of *HBG2* and *HBG1* that is associated with HPFH (Table 2). Edited progenitors derived from HUDEP-2 cells expressing HbA gave rise to ~50% F-cells and increased the *HBG/HBB+HBG* mRNA ratio; 58% of the erythroid cells derived from edited CD34^+^ progenitors were F-cells; HbF was ~20%. Erythrocytes derived from sickle CD34^+^ progenitors were 90% F-cells and protected from hypoxia-induced sickling [30,45]. 

CRISPR-Cas9 editing in both HUDEP-2 cells and CD34^+^ progenitors from patients with sickle cell anemia generated small deletions disrupting the binding motifs for: ZBTB7A 195–197 bp upstream of both *HBG2* and *HBG1*; the BCL11A binding motif 115 bp upstream of both *HBG2* and *HBG1*; and the QTL 158 bp upstream of *HBG2*. These induced mutations mimicked HPFH mutations (Table 2). The editing efficiency was >80%. Editing the ZBTB7A binding site in patient CD34^+^ cells resulted in: γ-globin transcripts forming ~50% of all non-α transcripts; ~80 F-cells; HbF comprising ~50% of hemoglobin tetramers; and ~70 of cells protected from hypoxia-induced sickling in vitro [30]. 

## 5. Conclusions 

Unusually high HbF levels of 20.7 ± 8.2% were present in ~5% of untreated patients with sickle cell anemia. When compared with patients with more customary HbF levels (3.1 ± 1.5%) their hemoglobin concentrations were higher (9.6 ± 1.5 vs. 8.1 ± 1.1 g/dL) and they were younger (15.2 ± 10.9 vs. 19.1± 10.8), although well beyond the age that HbF stabilizes in African Americans with sickle cell anemia [2,34]. Rarely are these unusually high HbF levels a result of mutations in the *HBB* gene cluster; perhaps more often they are caused by partial reversal of HbF gene repression by the minor alleles of the three known HbF-associated QTL. Nevertheless, QTL associated with HbF appear to account for less than half of HbF variance [46] and most instances of very high HbF remain unexplained. These instances could be a result of rare alleles of still uncharacterized HbF repressors or in genes, such as *KLF1*, that activate *BCL11A*, *ZBTB7A* and *HBB*. Variations in epigenetic HbF regulators, the intensity of stress erythropoiesis and cell stress signaling and a possible X-linked regulatory locus require further study [47,48,49,50,51,52]. With the widespread use of hydroxyurea to induce increased levels of HbF, it could be difficult to acquire a sufficient number of untreated cases to study. In addition to careful longitudinal clinical and hematologic examination, extensive genomic and transcriptomic analysis might help define previously uncharacterized HbF modulators.

## Figures and Tables

**Figure 1 jcm-09-03782-f001:**
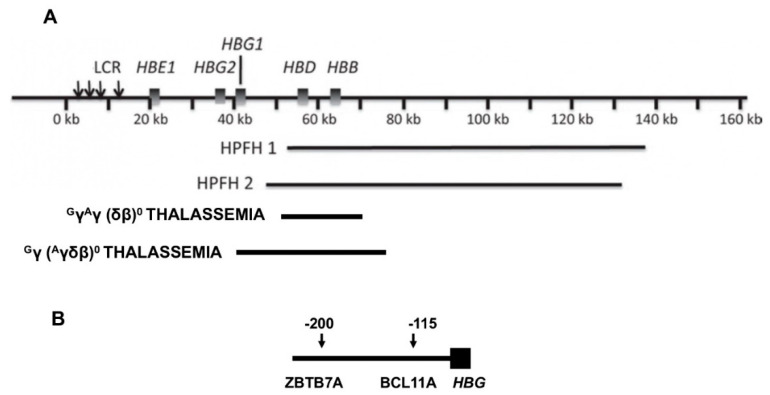
Deletions in the *HBB* gene cluster associated with increased HbF. (**A**) Displayed are the genes in the *HBB* gene cluster. The locus control region (LCR) that is upstream of *HBE* forms contacts with the promoters of the genes in this cluster affecting their transcription. When *HBB* and *HBD* are deleted, interactions between the promoters of the γ-globin genes and the LCR are favored. Beneath the *HBB* gene cluster are shown the locations and approximate sizes of common HPFH and δβ thalassemia deletions. (**B**) At positions −115 and −200 upstream of both *HBG2* and *HBG1* are binding motifs for the γ-globin gene repressors, ZBTB7A at −200 bp and BCL11A at −115 bp, respectively. Point mutations and small deletions in these regions and in nearby sites in the promoters, as detailed in Table 2, cause HPFH by altering the binding of these and other transcription factors.

**Table 1 jcm-09-03782-t001:** Hematologic findings in sickle cell disease with unusually high fetal hemoglobin (HbF).

	HbF (%)	Hemoglobin (g/dL)	Reticulocytes (%)	MCV (fL)
HbS-HPFH ^1^	32.9	12.6	1.7	74.9
HbS-δβ thalassemia ^2^	20–25	11	2	75–90
QTL ^3^	~15	9	-	80–100
Promoter mutations	Highly variable HbF; few reports in sickle cell anemia (see Table 2)			

^1^ Hereditary persistence of fetal hemoglobin (HPFH) 1, 2, 3 defined by gap-PCR. Results are averages in 51 cases, not adjusted for age. HbF was measured by different methods. ^2^ Few cases reported. ^3^ Sickle hemoglobin gene (HbS) homozygotes with 1 to 4 minor alleles of the 3 known quantitative trait loci (QTL) that are associated with HbF, as reported in [34].

**Table 2 jcm-09-03782-t002:** Point mutations in *HBG2* and *HBG1* promoters and the polymorphism in the *HBG2* promoter that have been associated with increased HbF in heterozygotes and compound heterozygotes with HbS. Approximate HbF levels are presented.

Mutation	Gene	HbF (%); Biology; Association with HbS
−567 T > G	*HBG2*	6–10: affects GATA and TAL1-binding motif
−4 bp (−225/−222)	*HBG1*	7
−211 C > T	*HBG1*	3–6
−202 C > G	*HBG2*	15–20; ZBTB7A binding site; associated with HbS trait
−202 C > T	*HBG1*	3 in sickle cell anemia: ZBTB7A binding site;
−202 + C	*HBG2*	25 in heterozygotes, 50 in homozygotes
−198 T > C	*HBG1*	4–12: ZBTB7A binding site
−196 C > T	*HBG1*	15–21: ZBTB7A binding site; 40 with β thalassemia
−197 C > T	*HBG1*	6: ZBTB7A binding site
−195 C > G	*HBG1*	5: ZBTB7A binding site
−175 T > C	*HBG1*	17–38: TAL1 binding site. 40 in HbS trait
−175 T > C	*HBG2*	28: TAL1 binding site
−161 G > A	*HBG2*	1–2
−158 C > T ^1^	*HBG2*	<1: higher levels under stress erythropoiesis; disruption of binding site increases HbS. The transcription factor binding here is uncharacterized.
−158 C > T	*HBG1*	3–5: occurs in cis to the −158 C > T in *HBG2*
−117 G > A	*HBG1*	9: BCL11A binding site
−114 C > T	*HBG2*	11–14: BCL11A binding site
−114 C > T	*HBG1*	3–6: BCL11A binding site
−114 C > G	*HBG2*	9: BCL11A binding site
−114 C > G	*HBG1*	38 in HbS heterozygote: BCL11A binding site
−13 bp (−114/−102)	*HBG1*	30: BCL11A binding site; associated with HbS
−113 A > G	*HBG1*	7: creates new GATA binding site
−110 A > C	*HBG2*	1: 3 in β thalassemia trait

^1^ Only the −158 *HBG2* C > T variant, rs7482144, is polymorphic. (www.globon.cse.psu.edu/) [29,35,36].

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
