# Peer review of "Fetal Hemoglobin in Sickle Hemoglobinopathies: High HbF Genotypes and Phenotypes"

_jcm, 2020, doi:10.3390/jcm9113782_

Round 1

Reviewer 1 Report

In this review, the author address the clinical implications of unusually high HbF levels in patients of African ancestry with sickle cell anemia, focusing on the HPFH phenotype due to  HBD/HBB gene deletions or point mutations in the promoters of the HbF genes.

I have only some minor points regarding this manuscript.

Line 60: “Reports of the genotypes of these QTL in HPFH have not been published”. Some reports have been published regarding HPFH in healthy individuals, including: Menzel S et al. (2007) Nat Genet 39:1197–1199 or Uda et al. PNAS, 2008 105 (5) 1620-1625.

Line 100: Change “MVC” by “MCV”

Line 143: Change “Table 1” by “Table 2”

Line 161: In table 1, Change “Hemoglobin (d/dL)” by “Hemoglobin (g/dL)”

Lines 170-178: Letters A and B were not detailed in the legend of figure 1.

Line 222: Change “Table 1” by “Table 2”

Lines 230-232: The author describes results on the CRISPR-Cas9  edition of binding motifs for ZBTB7A. However, data on the edition of BCL11A binding motif and the QTL -158 bp were not addressed.

Author Response

I have made the suggested changes. My remarks on line 60 refer only to the genotypes of these QTL in people with HPFH  genotypes and not healthy people as Menzel and others reported. I tried to clarify this point.

Reviewer 2 Report

This comprehensive and well-written review by Dr. Steinberg, explains in elegant detail, some of the known fundamental, as well as the confounding issues of high HbF genotypes and phenotypes in SCD. The review is detailed and comprehensive, and draws from the most recent findings, showing a highly commendable familiarity with the current literature. This is informative not just for scientists who have been in the field for some time, but for new upcoming investigators/trainees as well, as it helps navigate through important issues and the potential challenges of conducting formal studies (in the settings of widespread use of hydroxyurea).

I only have a few very minor comments.

  1. Sickle cell ANEMIA appeared to be used interchangeably with sickle cell DISEASE. Would be helpful for Dr. Steinberg to make it clear to the audience if a distinction exists.
  2. Line 63 "precursor" should be in the plural form
  3. Line 89 "most all". I am assuming a word is missing?

Author Response

Thanks. Corrections made